# Finding the Right Heavy Chains for Immunostimulatory Antibodies

**DOI:** 10.3390/ijms231810367

**Published:** 2022-09-08

**Authors:** Pierre Boulard, Valérie Gouilleux-Gruart, Hervé Watier

**Affiliations:** 1EA7501, GICC, Faculté de Médecine, Université de Tours, F-37032 Tours, France; 2Laboratoire d’Immunologie, CHU de Tours, F-37032 Tours, France

**Keywords:** immunostimulatory antibodies, Fc variants, antibody engineering, immunoglobulin heavy chains, therapeutic antibodies, Ge nomenclature

## Abstract

For twelve years, the oncology field has been revolutionized by antibodies targeting immune checkpoints. They must be considered as a heterogenous family of immunostimulatory antibodies displaying very different mechanisms of action, not only depending on the target or on the cells expressing it, but also on the IgG subclass or IgG variant that has been chosen. To dissect this complex landscape, the clinical experience has been confronted with a precise analysis of the heavy chain isotypes, referred as new Ge nomenclature. For antibodies targeting inhibitory receptors, anti-CTLA-4 antibodies (whose main effect is to kill regulatory T cells) will be distinguished from anti-PD-1 antibodies and other true antagonistic antibodies. Antibodies targeting ligands of inhibitory receptors (PD-L1, CD47) represent another different category, due to the antigen expression on tumors and a possible beneficial killing effect. The case of agonistic antibodies targeting lymphocyte activatory receptors, such as CD40 or 4-1BB, is still another “under construction” category because these products are less advanced in their clinical development. Altogether, it appears that choosing the right heavy chain is crucial to obtain the desired pharmacological effect in patients.

## 1. Introduction: The Perimeter of Immunostimulatory Antibodies

The scientific breakthrough and incredible medical success of antibodies targeting immune checkpoints in oncology add to the many other successes of anti-cancer antibodies, i.e., cytotoxic antitumor antibodies, anti-proliferative antibodies, anti-angiogenic antibodies, antibody-drug conjugates, bispecific T cell engagers and other bispecific antibodies. There is no appropriate term to encompass all or even part of the antibody-based therapeutic strategies, with the term immunotherapy being particularly difficult to handle [1] including for antibodies targeting immune checkpoints. In fact, the latter induce active non-specific immunotherapy [1], whereas antibodies usually determine specific passive immunotherapy. This particular point can however be nuanced by the fact that different preclinical studies using murine models and different clinical data have shown that classical antibodies directed against the tumor can actively induce memory T cell responses [2].

We therefore argue for the term immunostimulatory antibodies to encompass antibodies designed to wake up the immune system and trigger antitumor immunity. This term has the advantage to include several categories of antibodies, those that target immune checkpoints and those that directly activate cells of the immune system. Immune checkpoints are lymphocyte inhibitory receptors such as CTLA-4 (cytotoxic T-lymphocyte-associated protein 4), PD-1 (programmed cell death 1), etc., whose primary function is to switch off immune effector cells, but they usually include their specific membrane ligands, such as PD-L1 (programmed death-ligand 1). Many antibodies against immune checkpoints are already approved while antibodies of the second category, which directly activate immune effector cells, are still under clinical development, probably because it is not so easy to control activatory receptors such as CD40, CD137, etc. This review will focus on antibodies that are marketed or at least in phase 3, in order to have both reliable molecular information about the product and information about efficacy in patients. Indeed, the sequence is publicly disclosed when the international non-proprietary name (INN) is attributed by the WHO, i.e., when the product is well advanced. Our analysis was done after screening the scientific literature as well as the TABS [3], IMGT [4], WHO/INN [5] and clinical trials [6] databases. A few antibodies in early clinical developments were added when they explored new pharmacological properties, without trying to have a comprehensive catalogue of possible immune targets. To lighten the text below, antibodies are only referred to by their INN, where applicable.

## 2. Playing with IgG Subclasses to Adjust the Pharmacological Effects

Trying to distinguish the different classes of immunostimulatory antibodies is more than a simple conceptual or semantic exercise. Pharmacological effects clearly differ from one category to another due to the nature of the target, which could be an activatory or an inhibitory receptor. Moreover, the choice of the IgG heavy chain isotype determines the Fc region- and hinge-associated properties, which are of utmost pharmacological significance. In oncology, it is common to choose the γ1 isotype (IgG1 subclass) to favor the potential of tumor destruction through the recruitment of immune effectors. The latter could be humoral with the activation of the complement cascade, or cellular with cytotoxic or phagocytic cells expressing receptors for the Fc region of IgG or FcγR. IgG3 (γ3 isotype) is another highly potent subclass in terms of immune effector activation, but it has been discarded due to its reduced half-life. By contrast, IgG2 (γ2 isotype) and IgG4 (γ4 isotype) are also avoided to kill cancer cells because they have a lower ability to recruit immune effectors. The most visible structural differences between IgG1 and IgG2/IgG4 are located in the hinge region, as shown in Figure 1.

The case of cetuximab and panitumumab, two anti-EGFR (epidermal growth factor receptor) antagonistic antibodies, is a remarkable example. They do not really differ in their ability to antagonize EGFR, but the former is an IgG1 and the latter is an IgG2. Both are effective in colorectal cancer without KRAS mutation, suggesting that antagonizing EGFR is sufficient to achieve the therapeutic effect. In contrast, only cetuximab has been shown to be effective in head and neck cancers, panitumumab having proved to be clinically inefficient [7]. In this indication, antagonization of EGFR is probably not sufficient and the cytolytic action of IgG1 through the recruitment of immune effectors appears to be necessary to obtain an antitumor activity.

In reality, the distinction between cytolytic IgG1 and non-cytolytic IgG2/IgG4 is much more complex. On one hand, IgG2 and IgG4 are not totally silent in terms of either complement activation or FcγR-positive effector cell recruitment. In particular, IgG4 retains the property of binding to some FcγR and to exert targeted cell killing, at least in some patients [8]. On the other hand, IgG1 do not necessarily prove to be cytolytic in patients, which is well illustrated by some antagonistic antibodies such as efalizumab (anti-integrin LFA-1 [lymphocyte function-associated antigen 1]) which do not induce lymphopenia. In addition, advances in antibody engineering now make it possible to obtain IgG1 whose cytotoxic effect is boosted (boosted IgG1), or on the contrary totally abolished (IgG1 with silent Fc region or silent IgG1), or IgG4 having completely silent Fc regions (silent IgG4). As discussed further, other variations can be introduced to improve in vitro (storage, pre-administration) or in vivo antibody stability. The increasing number of engineering variants greatly expands the range of pharmacological potentialities. To identify them more easily, we have proposed a new nomenclature system called Ge (IgG engineered), sub-divided in G1e, G2e and G4e according to the subclass, inspired by the Gm allotype nomenclature system (IgG marker) [9]. Table 1 (G1e variants of IgG1) and 2 (G4e variants of IgG4) show an updated version engineered mutations present in marketed antibodies and Fc-fusion proteins. The diversity of antigenic targets of immunostimulatory antibodies combined with the diversity of IgG subclasses and their variants induces a complex landscape, which is however quite informative concerning the pharmacology of these drugs in Human.

## 3. Targeting of Inhibitory Receptors (Immune Checkpoints) or Their Ligands

The term “immune checkpoint” came of age in 2018 when the Nobel Prize was attributed to Prof. James Allison and Prof. Tasuku Honjo for the discoveries of inhibitory functions of CTLA-4 and PD-1, respectively. These immune checkpoints play a harmful role in cancer by curbing the immune system and favoring tumor progression. Therefore, their blockade by antibodies became a smart concept to fight cancer, following the idea that the enemies of my enemies are my friends. The concept became more than clinically valid when the first anti-CTLA-4 was approved in 2011 and the first anti-PD-1 in 2014. The term “immune checkpoint inhibitor” (ICI) is now well popularized and very commonly used, although the pharmacological reality is much more complex. This is particularly true for anti-CTLA-4 antibodies that will be analyzed first in this chapter, before anti-PD-1 antibodies, which are true ICIs while still raising interesting questions. Next, we will discuss the choice of IgG subclass for targeting LAG-3 and TIGIT. Finally, we will discuss the anti-PD-L1 and anti-CD47 antibodies, which are not tumor-expressed ligands of immune checkpoints, and which mandate a different reasoning from that underlying the targeting of checkpoints expressed on immune cells.

### 3.1. Anti-CTLA-4 Antibodies

With hindsight, the case of anti-CTLA-4 (CD152) antibodies is very instructive and illustrates the difficulty in predicting the activity of an antibody subclass in Human. In 2007, two antibodies were in competition to reach market approval: ipilimumab, which is an IgG1κ (unmutated/G1e0 γ1 heavy chain) and tremelimumab, which is an IgG2κ (unmutated/G2e0 γ2 heavy chain). Choosing IgG2 with low effector functions was well in line with the expected pharmacological profile of a purely antagonistic antibody, preserving CTLA-4 expressing cells from lysis, and was then considered reassuring [10]. Conversely, choosing an IgG1 for ipilimumab was somewhat disturbing, as it could induce unintended effects, starting with the destruction of anti-tumor lymphocytes that we wanted to wake up. Preclinical trials of ipilimumab did not shown any safety concern [11]; paradoxically, ipilimumab obtained a first marketing authorization in 2007 for the treatment of melanoma, while tremelimumab is not yet approved, despite multiple phase 3 trials. Obviously, there is no proof that the inability of IgG2 to recruit immune effectors is responsible for the failure of tremelimumab, but the hypothesis seriously merits being raised. In other words, how can one explain that an immune checkpoint antagonist equipped with a depleting activity is more effective than a purely antagonistic anti-CTLA-4 antibody?

Experiments in mice provided the first level of explanation, thanks to an anti-murine CTLA-4 antibody available under different murine IgG formats [12]. In these models, the murine IgG2a antibody subclass (γ2a isotype, with depleting activity) showed anti-tumor activity, contrary to the same antibody of the murine IgG1 antibody subclass (γ1 isotype), known for its lower cytotoxic activity [12]. The difference was explained by the ability of anti-CTLA-4 mIgG2a but not anti-CTLA-4 mIgG1 to eliminate intra-tumoral regulatory T cells (Treg) [12]. Ingram et al. also showed that the IgG2a Fc was required for efficacy while using an anti-CTLA-4 high-affinity alpaca heavy chain-only antibody fragment [13]. A second level of demonstration was provided by Du et al. who observed in two different models of humanized mice that ipilimumab has a depleting effect on Tregs [14]. The third argument is provided by clinical observations in line with what was observed in murine models, where patients treated with ipilimumab have less Treg infiltration in their tumors compared to patients treated with tremelimumab [15,16]. However, this study presented several limits that have been discussed elsewhere [17] thus preventing a definitive conclusion. The fourth and most definitive argument relies on the very high variability of clinical response to ipilimumab, with a low proportion of responder patients. Under ipilimumab treatment, homozygous patients for the 158-valine (158V) alloform of FcγRIIIA (FcγRIIIA-158VV), whose tumor has a high neoantigenic burden, have a better survival rate than any other patients. FcγRIIIA-158VV homozygotes represent between 15 and 20% of the population and are always the best responders for patients when treated by cytotoxic antibodies. This has been now confirmed in many cohorts of patients since 2002 when we demonstrated it in a cohort of patients treated with rituximab [18]. On the contrary, patients carrying the 158-phenylalanine (158F) allotype of this receptor (heterozygotes or homozygotes) present a poorer response to ipilimumab, even when their tumors display a high neoantigenic burden, which is a prerequisite for anti-tumor T cell responses [19]. Not surprisingly, patients with a low neoantigenic burden do not respond to ipilimumab, whatever their FcγRIIIA genotype [19]. In the case of ipilimumab, which has now to be categorized as a cytolytic antibody—the role of FcγRIIIA is reinforced by the fact that the tumors of patients responding to ipilimumab show a greater infiltration of CD68+ macrophages [15] and CD56+ NK lymphocytes [20], the two effector cell populations that express FcγRIIIA. These results clearly indicate that the recruitment of effector cells by the Fc region of ipilimumab is critical for therapeutic efficacy, and that the efficacy of anti-CTLA-4 antibodies could be optimized, at least in patients with a high antigen burden and carrying the other FcγRIIIA-158F allotype which has a lower affinity for IgG1. Preclinical data already show that antibodies whose Fc region has been engineered to increase its affinity for FcγRIIIA, and their ability to recruit FcγR+ cells has a better therapeutic activity [21]. One such boosted antibody, botensilimab (AGEN1181), has ended its phase 1 conclusively with favorable tolerance [22,23] and should start its phase 2 trials shortly. It carries the S239D, A330L, I332E triple mutation described to specifically enhance its binding to FcγRIIIA, notably FcγRIIIA-158F [24,25].

To conclude about anti-CTLA-4 antibodies, it is now clear that these immunostimulatory antibodies are not simple ICIs. Du et al. have even demonstrated, quite unexpectedly, that ipilimumab does not exhibit antagonistic effect on CTLA-4 [14], an observation that should lead to the conclusion that it does not merit its denomination of immune checkpoint inhibitor! Their main mechanism of action being the depletion of Treg cells, a paradigm shift is taking place which opens up the pathway to the development of cytotoxic antibodies targeting Treg lymphocytes. This could be the case of anti-CD25 antibodies [26] and anti-OX40 (CD134) antibodies [27].

### 3.2. Anti-PD-1 Antibodies

Unlike anti-CTLA-4 antibodies, anti-PD-1 antibodies probably better satisfy the strict definition of ICI, since they are actually antagonizing an inhibitory receptor expressed by cancer effector lymphocytes. Murine models confirm that an anti-PD-1 (depleting) IgG2a does not present any antitumor activity, unlike a murine anti-PD-1 IgG1, or even a murine IgG1 devoid of effector functions [28]. Similarly, both anti-PD-1 antibodies of the human IgG4 subclass or equipped with a silent human IgG1 Fc region have good antitumor activity in FcγR humanized mice. Silent IgG1 are either obtained by producing an aglycosylated variant through mutation in the N297 glycosylation site (G1e8, G1e11, G1e18), or by mutating critical residues in the FcγR interaction site of IgG, notably L234 and L235 (G1e9, G1e10, G1e15). In fact, unanimously, all the marketed anti-PD-1 antibodies are devoid of effector functions, being either IgG4 (pembrolizumab, nivolumab, cemiplimab, dostarlimab, tislelizumab, toripalimab and sintilimab) or silent IgG1 (penpulimab, as well as budigalimab and prolgolimab that are in clinical development) [29,30]. The anti-PD-1 pidilizumab (IgG1κ, G1e0) never passed the phase 2, not necessarily because it had not been Fc region silenced but more probably due to a restricted reactivity to certain PD-1 glycoforms and cross-reactivity with DLL-1 (delta-like canonical Notch ligand 1). These seven IgG4 anti-PD-1 (pembrolizumab, nivolumab, cemiplimab, dostarlimab, tislelizumab, toripalimab and sintilimab) are not bona fide IgG4 (IgG4 G4e0) but all are engineered IgG4 with the S228P point mutation in the hinge region (G4e1, Table 2 and Figure 1). The S228P mutation does not affect the properties of the Fc region, but abolishes the ability of IgG4 to form hemi-IgG and to reassociate with other hemi-IgG4 to form bispecific antibodies (Fab-arm exchange) (Figure 2) [31]. This phenomenon has been demonstrated in patients treated with natalizumab [32], the only unmutated IgG4 (G4e0) that is used in the clinic. The Fab-arm exchange phenomenon (Figure 2) and also the in vitro instability linked to S228 could be both prevented by the S228P mutation, which is not protected by a patent [31]. This mutation has been used for a long time (Table 2), and is now considered as a standard for therapeutic IgG4, being highly recommended by regulation agencies in replacement of natural G4e0 IgG4 [31].

Whether or not carrying the S228P mutation, IgG4 retain the property of binding to FcγRI and FcγRIIIA-158V and are therefore not totally devoid of cytolytic potential [29]. Therefore, it could not be excluded that anti-PD1 IgG4 antibodies could deplete anti-tumor effector T cells, leading to the abolition of its therapeutic effects. It is indeed known that some anti-PD1 treated patients show paradoxical responses (hyperprogressors) [33]. The fact that whether or not these hyperprogressive patients are FcγRIIIA-158V homozygotes would really deserve to be investigated. To the best of our knowledge, this has never been done.

Preclinical models even suggest that we would gain in efficacy by fully abolishing the ability of anti-PD-1 IgG4 to recruit FcγR+ cells, notably those expressing FcγRI [29]. Tislelizumab is an anti-PD-1 IgG4 approved in China, whose Fc region has been largely modified. Part of the mutation is intended to reinforce the stability of the molecule (double S228P and R409K mutation) while the other to totally abolish the Fc region functions (Fc region silent, via a quadruple E233P, F234V, L235A and D265A mutations). Whether tislelizumab compares favorably with the other anti-PD1 IgG4 will require adequate clinical studies. Similarly, penpulimab is a silent IgG1 that could have a better clinical profile than the classical IgG4 G4e1 antibodies.

### 3.3. Anti-TIGIT and Anti-LAG-3 Antibodies: Antagonist Antibodies?

Lymphocyte-activation gene 3 (LAG-3) is a MHC class II inhibitory receptor expressed on many T-cell subpopulations [34] and one of the new immune checkpoints targeted by monoclonal antibodies. As with anti-PD1 antibodies, anti-LAG-3 antibodies have been developed using formats having low or no Fc effector functions. This is notably the case of relatlimab, which is now approved in association with nivolumab in the treatment of advanced melanoma [35]. As a S228P IgG4 (G4e1), its main mechanism of action consists in inhibiting the LAG-3-MHC class II interaction and activating exhausted T cells to restore antitumoral activity. The same G4e1 format has been chosen for favezelimab, miptenalimab, ieramilimab and encelimab, which are also anti-LAG-3 antibodies. Fianlimab differs from the previous anti-LAG-3 antibodies; it is an IgG4 that has been further mutated (E233P/L234V/G235A/S228P) to completely silence its Fc region [36]. Currently in phase 3, it will still require some time to know if it compares favorably with other anti-LAG-3 antibodies!

Like PD1 and LAG-3, TIGIT (T cell immunoreceptor with Ig and ITIM domains) is another immune checkpoint expressed on a diverse set of effector cells (NK, CD4 and CD8 T lymphocytes). Engaged by its tumor cell-expressed ligands, CD155 and, to a lesser extent, CD112, TIGIT induces negative signals [37]. However, very surprisingly, most anti-TIGIT under clinical development are unmutated IgG1 (G1e0), presenting cytotoxic activity, contrary to anti-PD1 and anti-LAG-3 antibodies. One of them, tiragolumab, recently failed in different phase 3 trials, either in monotherapy or association with atezolizumab. Many explanations could be provided to explain this failure, but we could reasonably hypothesize that an unmutated IgG1 was the wrong choice. It will now be interesting to know the results of phase 3 clinical trials of domvanalimab (G1e10), which has been designed to avoid the recruitment of effector cells and is therefore a pure antagonistic activity.

### 3.4. Anti-PD-L1 (and Anti-CD47) Antibodies

PD-L1, as the main ligand of PD-1, has been considered as an alternative target to PD-1 from the beginning of the PD-1 story, and many anti-PD-L1 antibodies have been approved. Similarly for PD-1, the preferred strategy was the simple blockade of PD-L1 with IgG4 (decreased effector functions) or Fc silent IgG1 (no effector functions). This was followed by four of the five marketed anti-PD-L1 antibodies: atezolizumab has a N297A mutation (G1e8) that abolishes the N-glycosylation site and effector functions [38]; durvalumab has a L234F/L235E/P331S triple mutation (G1e9) affecting the FcγR binding site [39,40]; sugemalimab is an unmutated IgG4 (G4e0) only approved in China; and envafolimab is a camel-derived VH region fused to a human IgG1 Fc region with a C220S/D265A/P331G triple mutation, two of them (D265A/P331G) directly affecting binding to FcγR [41].

The clinical success of these anti-PD-L1 antibodies was not predicted by preclinical models. Indeed, it was evidenced that murine IgG2a anti-PD-L1 antibodies with cytolytic properties have a higher therapeutic efficacy than its murine IgG1 equivalent [28]. The benefit of adding immune effector function could be explained by the fact that PD-L1 is expressed on tumor cells, and that it could be advantageous to kill tumor cells in addition to PD-L1 blockade. The only marketed Fc region-competent anti-PD-L1 antibody is avelumab (IgG1 G1e0), which is indicated in Merkel cell carcinoma, where its cytotoxic activity is undoubtedly essential [42], and in advanced renal cell carcinoma (in association with axitinib). However, due to a lack of direct comparison between avelumab with any of the other anti-PD-L1 antibodies [43,44], it remains unclear whether Fc region function are desirable or not in Human. Therefore, it appears quite premature to go one step further in thinking to anti-PD-L1 with boosted IgG1, although a biotech company has already developed a hypofucosylated anti-PD-L1 antibody (G1e4), with increased binding to FcγRIIIA [45].

Similar questions can be raised for drugs targeting CD47, an antigen overexpressed on cancer cells. CD47 induces a “don’t eat me” signal to macrophages expressing SIRPα, an immune checkpoint expressed on myeloid cells. In this particular situation, the objective is to combine a CD47-blocking effect to restore the phagocytic capacity with an opsonization effect with Fc-competent molecules. IgG1 are more potent than IgG4, but IgG4 remain functional since macrophages do express FcγRI. The most advanced product is magrolimab, an anti-CD47 G4e1 IgG4, which is in phase 3 in acute myeloid leukemias and myelodysplastic syndromes. The case of TTI-621 (SIRPα-IgG1 Fc region) and TTI-622 (SIRPα-IgG4 Fc region) is probably more instructive, due to the fact that both molecules have entered into clinical development and provide a very rare opportunity to compare the pharmacological effect of different Fc region, combined to the same antigen-binding element. TTI-621 appears as more potent than TTI-622 in preclinical models, whereas a totally silent SIRPα-mutated IgG4 Fc region was less potent that TTI-622 [46]. However, some safety issues were observed in TTI-621 clinical trials, due to the wide expression of CD47 on normal blood cells [47]. Whether TTI-622 will induce less thrombocytopenias while keeping clinical efficacy now remains to be determined in large clinical trials.

## 4. Agonistic Antibodies Targeting Lymphocyte Activation Receptors

A totally different approach for immunostimulatory antibodies is to engage lymphocyte receptors to induce activating signals and enhance antitumor activity. This approach has been mainly developed by targeting members of the TNF-α receptor family. The use of such agonistic antibodies in oncology is an old idea that was initially developed to induce tumor apoptosis by targeting death receptors, notably the two TRAIL receptors (tumor-necrosis-factor-related apoptosis-inducing ligand), DR4 (mapatumumab) and DR5 (lexatumumab, tigatuzumab, conatumumab, drozitumab). All were IgG1 to potentiate the cytolytic effect, and some studies have shown that the co-engagement of FcγR was necessary for their proapoptotic effect [48].

In the case of immunostimulatory agonistic antibodies, the cytolytic effect must be avoided to preserve the cells to be activated. Many targets have been identified, CD40 and 4-1BB (CD137), but also OX40 (CD134), CD27, GITR (CD357, glucocorticoid induced TNFR family-related gene), as well as receptors not belonging to the TNFR family such as ICOS (CD278, inducible T-cell costimulatory). Induction of an agonistic signal hardly depends on the epitope recognized on the receptor [49]. It also depends on the ability of an antibody to aggregate two receptor molecules, and even on the capacity of the antibody to co-engage FcγR, not at all to stimulate effector functions but to induce an additional level of aggregation (clustering) of targeted receptors and thereby an increase in agonist signal strength. In particular, murine models have highlighted the interest of using murine IgG1 rather than IgG2a, as well as the importance of the FcγRIIB inhibitory receptor. However, the transposition into the clinic is far from being easy; no immunostimulatory antibodies of agonist-type are approved yet. In all cases, great caution is required because of the risk of inducing a massive cytokine release during infusions. This was notably the case with TGN1412, an anti-CD28 superagonistic IgG4, which triggered a cytokine storm in healthy volunteers. This disastrous trial nevertheless made it possible to investigate further the underlying mechanisms, showing that CD28 aggregation induced by TGN1412 was facilitated by FcγRIIB human co-engagement, although it is an IgG4 antibody [50].

### 4.1. Anti-CD40 Antibodies

Selicrelumab (CP-870,893) was the first anti-CD40 agonistic antibody tested in the clinic, showing limited efficacy in pancreatic cancer [51]. This antibody was selected for its high agonistic activity. IgG2 was not a deliberate choice because selicrelumab is derived from transgenic mice producing human IgG2 but it secondarily appeared that the IgG2 subclass favors the agonistic effect [52]. In fact, IgG2 hinge region is shorter than that of IgG1 and comprises two additional cysteines allowing the formation of two more disulfide bridges (Figure 1), making IgG2 very rigid, especially when it adopts the IgG2B isoform [52]. This rigidity accentuates the aggregation of the receptor, and consequently, the agonistic effect. In models of transgenic mice expressing CD40 and human FcγR, a variant of selicrelumab whose Fc region has an increased affinity for FcγRIIB, but not FcγRIIA showed a more potent antitumor effect, but at the cost of increased toxicity [53]. Indeed, the use of selicrelumab was accompanied by cytokine release syndromes, as well as platelet and liver toxicities. Apart from the CDX-1140 antibody, which is also an IgG2, the other biopharmaceutical companies developing anti-CD40 agonistic antibodies have made other choices: unmutated IgG1 Fc region for dacetuzumab, mitazalimab and lucatumumab, silent IgG1 (V273E) for giloralimab and S267V-mutated IgG1 for sotigalimab, this latter mutation enhancing the binding to FcγRIIA and FcγRIIB. Mitazalimab is still in phase 2 while dacetuzumab and lucatumumab were discontinued due to their modest antitumor activities. Moreover, it could be observed that the response to dacetuzumab did not depend on FcγRIIA and FcγRIIIA polymorphisms [54], suggesting an absence of cytolytic effect for this unmutated IgG1. The choice of an enhanced binding to FcγRIIA and FcγRIIB for sotigalimab is quite unexpected given the safety issues already observed selicrelumab. Giloralimab has now entered a phase 2 trial that should answer the question of the clinical interest of silencing the Fc region.

### 4.2. Anti-4-1BB Antibodies

Urelumab and utomilumab are two agonistic antibodies directed against the 4-1BB activator molecule. They differ both by their binding to 4-1BB [55] and by their heavy chain isotype, which makes any comparison difficult. Urelumab is a purely agonistic IgG4 G4e1 (S228P) (does not block ligand binding) while utomilumab is an IgG2 dual agonist and antagonist, blocking endogenous ligand binding while activating 4-1BB-expressing cells. During clinical trials, urelumab (strong agonist, IgG4 G4e1) was significantly more toxic (liver toxicity) than utomilumab (weak agonist, IgG2), which did not prove to be very effective. Therefore, this observation does not argue in favor of IgG2 always being more agonistic and more at risk of toxicity; however, there are too many differences between the two products to draw a solid conclusion. LVGN6051 is a new anti-4-1BB antibody designed as a weak agonist (like utomilumab), but whose Fc region has been engineered to increase its binding to FcγRIIB, but the mutation has not been disclosed [56]. Phase I clinical trial demonstrated a satisfying safety profile and some antitumor activity that requires further evaluation [57].

The great perplexity that reigns over the choice of the heavy chain isotype for agonistic anti-CD40 and anti-4-1BB antibodies also applies for other targets, as shown for anti-OX-40 antibodies [58].

## 5. Conclusions

The rationale underlying the choice IgG subclasses or their variants for immunostimulatory antibodies is schematically depicted and summarized in Figure 3.

This rational was built up en route, at the favor of many clinical developments, with a lot of fumbling. Animal models proved to be little predictive of therapeutic results, notably because FcγR highly differs between Human and other mammals [59]. It is now quite easy and straightforward to design receptor antagonists for targeting inhibitory receptors like PD-1 or LAG-3, but the trend is now to develop pure antagonists, with totally silent IgG, devoid of any residual cytotoxic activity. The situation becomes more complicated in case of target antigens expressed on Treg lymphocytes (anti-CTLA-4) or on tumor cells (anti-PD-L1, anti-CD47), where it could be of high interest to have IgG equipped with effector functions and possibly boosted IgG1 to kill the cells expressing the targeted antigen. The situation is even more complex for agonistic antibodies targeting activatory receptors, for which the choice of epitope, hinge and Fc region are crucial to obtain a delicate balance between efficacy and undesirable effects. For the latter, only the future will tell us which should be the best choice!

Isotype engineering applied to immunostimulatory antibodies is still in its infancy. One could imagine a finer tuning of the Fc region to increase the binding to some FcγR and not others. In immunostimulatory antibodies, it is also surprising that none of the mutations described and clinically used to increase the binding to FcRn at pH6 and to increase the half-life of IgG [60] have not been introduced in the Fc region. This could be because these drugs are very potent immunostimulants and that the reasonable choice is not to have an extended duration of action in case of adverse effects. Finally, isotype engineering will be necessary to create bispecific antibodies, notably to focus their action on particular immune cells.

## Figures and Tables

**Figure 1 ijms-23-10367-f001:**
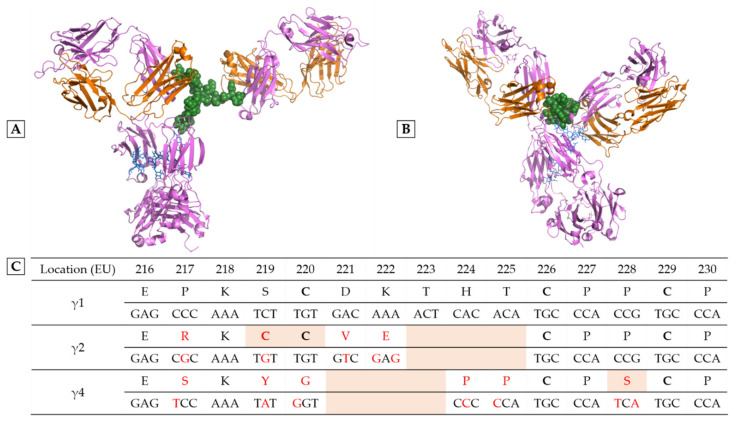
Differences between human IgG subclasses in the hinge region. Structure of an IgG1 (**A**) and an IgG4 (**B**), the only two human IgG subclasses that have been crystallized in their entirety. The heavy chains are colored in pink, the light chains in orange; N-glycans are indicated in blue, and the hinge regions in green. Figures were elaborated using PyMOL Molecular Graphics System, version 1.7.4 (Schrödinger) from PDB file 1HZH representing the unique human IgG1 crystallized, named B12 and directed against the gp120 of HIV (human immunodeficiency virus) and from PDB file 5DK3 representing pembrolizumab, the only human IgG4 crystallized (G4e1 variant). The high flexibility of the IgG1 hinge region gives freedom to the Fab arms and causes strong asymmetry of the whole molecule. Unfortunately, there is no available structure of an entire human IgG2. (**C**) Hinge region alignment of heavy chains γ1, γ2, γ4. Amino acids (according to EU numbering) and nucleotides that differ to IgG1 sequence are colored in red. Cysteines engaged in disulfide bridge are in bold. Notable differences appear in orange background: 3 amino acids deletion in IgG2 and IgG4, cysteines 219 and 220 of IgG2, and serine 228 of IgG4 (see text).

**Figure 2 ijms-23-10367-f002:**
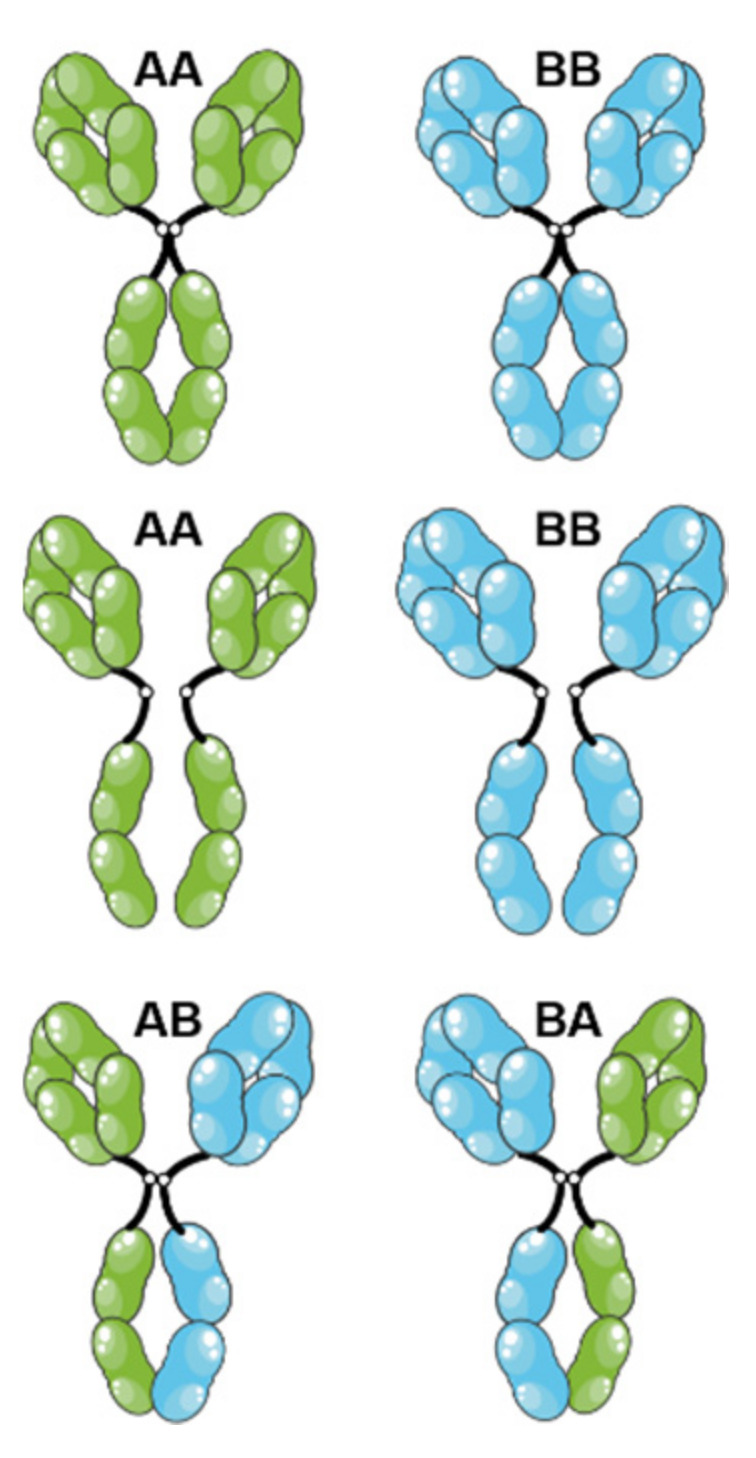
Fab-arm exchange of IgG4. Each IgG4 could dissociate in two parts at the level of the hinge region and of the Fc region, resulting in two half-IgG4 (one heavy chain associated with its light chain). These half-IgG4 could reassociate with any other half-IgG4, creating bispecific antibodies, less avid for the antigen and less prone to create immune complexes. This is a natural property of IgG4, not necessarily desirable for therapeutic purposes. The figure was partly generated using Servier Medical Art, provided by Servier, licensed under a Creative Commons Attribution 3.0 unported license.

**Figure 3 ijms-23-10367-f003:**
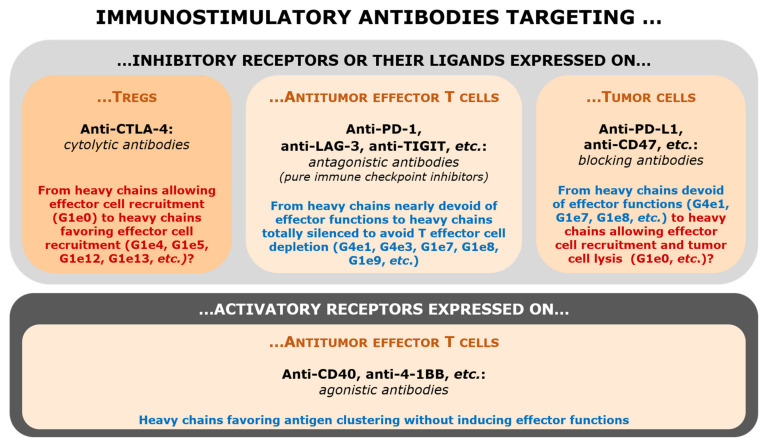
Immunostimulatory antibodies overview.

**Table 1 ijms-23-10367-t001:** Updated version of the nomenclature of G1e variants of IgG1, with a focus on silent (blue highlighting ^1^) and boosted antibodies (red highlighting ^1^), and immunostimulatory antibodies highlighted in capital letters.

Ge Numbering	Heavy Chain Variations (EU Numbering) ^2^	INN (Year of First Approval in the EU, USA or JP)
**G1e0**	No variation	Many examples, notably: **IPILIMUMAB** (2011), **AVELUMAB** (2017)
**G1e1**	Lacks CH1 + C220S C226S C229S P238S substitutions	abatacept (2005), belatacept (2011)
**G1e2**	Lacks CH1 + Production in *E.coli* (aglycosylation)	romiplostim (2008)
**G1e3**	Lack CH1 + Deletion of the five first amino acids of the hinge region	aflibercept (2011), efmoroctocog α (2014), eftrenonacog α (2014)
**G1e4**	Hypo- or afucosylation	mogamulizumab (2012), benralizumab (2017), naxitamab (2021), inebilizumab (2022)
**G1e5**	Addition of a bisecting GlcNAc also indirectly inducing hypo- or afucosylation	obinutuzumab (2013)
**G1e6**	F126L substitution	ramucirumab (2014), necitumumab (2015)
**G1e7**	L235A G237A substitutions	vedolizumab (2014)
**G1e8**	N297A substitution	**ATEZOLIZUMAB** (2016)
**G1e9**	L234F L235E P331S substitutions	**DURVALUMAB** (2017)
**G1e10**	L234A L235A substitutions	etesevimab (2021), risankizumab (2019)
**G1e11**	K213A N297A substitutions	eptinezumab (2020)
**G1e12**	S239D K274Q Y296F Y300F L309V I332E A339T V397M substitutions	tafasitamab (2020)
**G1e13**	L235V F243L R292P Y300L P396L substitutions	margetuximab (2020)
**G1e14**	F405L substitution and afucosylation	amivantamab chain A (2021)
	K409R substitution and afucosylation	amivantamab chain B (2021)
**G1e15**	L234F L235E M252Y S254T T256E P331S substitutions	tixagevimab/cilgavimab (2021)
**G1e16**	M428L N434S substitutions	sotrovimab (2021)
**G1e17**	M252Y S254T T256E H433K N434F susbtitutions	efgartigimod (2021)
**G1e18**	N297G T366W substitutions (chain A)	mosunetuzumab (2022)
	N297G T366S L368A Y407V substitutions (chain B)	

^1^ Variants devoid of effector functions are highlighted in blue, those boosted for their effector functions in red, those modified for increased half-life in yellow, those with both increased half-life and decreased effector functions therefore in green, those with not interfering Fc functions in grey. ^2^ The deliberated deletion of the C-terminal lysine (K447) is not included in this nomenclature.

**Table 2 ijms-23-10367-t002:** Updated version of the nomenclature of G4e variants of IgG4, with a focus on silent (blue highlighting ^1^) and immunostimulatory antibodies highlighted in capital letters.

Ge Numbering	Heavy Chain Variations (EU Numbering) ^2^	INN (Year of First Approval in the EU, USA or JP)
**G4e0**	No variation	natalizumab (2004)
**G4e1**	S228P substitution	gemtuzumab ozogamicin (2000), **PEMBROLIZUMAB** (2014), **NIVOLUMAB** (2014), **CEMIPLIMAB** (2018), etc.
**G4e2**	Hybrid IgG2 (up to T260)/IgG4	eculizumab (2017)
**G4e3**	S228P, F234A and L235A substitutions	dulaglutide (2014), galcanezumab (2018)
**G4e4**	K196Q, S228P, F296Y, K439E and L445P substitutions plus removal of G446	emicizumab chain A (2017)
	K196Q, S228P, F296Y, E356K, H435R and L445P substitutions plus removal of G446	emicizumab chain B (2017)
**G4e5**	Hybrid IgG2 (before T260)/IgG4 (after) and M428L, N434S substitutions	ravulizumab (2018)
**G4e6**	S228P, L235E substitutions	**SUTIMLIMAB** (2022)

^1^ Variants devoid of effector functions are highlighted in blue, those with both increased half-life and decreased effector functions in green and those with not interfering Fc functions in grey. ^2^ The deliberated deletion of the C-terminal lysine (K447) is not included in this nomenclature.

## Data Availability

Not applicable.

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
