# Peer review of "Finding the Right Heavy Chains for Immunostimulatory Antibodies"

_ijms, 2022, doi:10.3390/ijms231810367_

Round 1

Reviewer 1 Report

Revision of the review : “Finding the right heavy chains for immunostimulatory antibodies”.

The review is divided in a total of five chapters, namely a first introductory part defining the immunostimulatory antibodies, followed by a second chapter describing the choice of the different type of IgGs to modulate the pharmacological effects of the diverse antibodies. The third part is about the antibodies tested in clinical trials or already approved in clinics that target inhibitory receptor or their ligands (e.g. CTL4, PD-1, PD-L1). The fourth chapter is about another type of antibodies, agonistic antibodies, used to target activation receptors of the lymphocyte. The review finishes with a last chapter, the conclusion.

The review is well written and I consider that it is of interest for the scientific community and I would recommend it for publication. However, the minor revisions hereafter should be addressed.

Minor considerations.

-   Table 1: it would be interesting to include in the table the impact of those variations and thus helping the reader to follow it.

-   Line 219 : Please include some references.

I would recommend that the authors include some figures to help the reader to better follow the review. For instance, a figure with the inhibitory receptor and/or their ligands (e.g. CTL4, PD-1, PD-L1) expressed in the respective cells with the respective targeting antibodies. Another figure, for the agonistic antibodies would also be helpful.

Reviewer 2 Report

Review for “Finding the right heavy chains for immunostimulatory antibodies” by Boulard et al.

In this review manuscript the authors present a discussion of the rationales and findings that go into choosing the best type of heavy chains to use in immunostimulatory antibodies. The authors begin by covering various mechanistic aspects of what makes up an immunostimulatory antibody. They next delve into examples of therapeutically applied immunostimulatory antibodies and how different examples benefitted from the use of different varieties of heavy chains. Finally, the authors consider agonistic antibodies, which present a new set of questions relative to past findings with approved immunostimulatory antibodies.

The manuscript covers an important area of basic biomedical and clinical research. The nomenclature used in the field is not straightforward and the authors do a commendable job of devising systematic methods of describing heavy chains. The senior author is an established contributor to this field and this work builds on previous contributions (such as reference #9).

This review will be a valuable contribution to the field. There are a few points I feel the authors should address before publication.

*This manuscript is primarily about developing and describing the Ge nomenclature system; however, the Gm nomenclature system is also referenced and used at some places throughout the paper. If Gm nomenclature is to be used alongside Ge nomenclature the authors should devote some text to making clear how Gm naming is different than Ge naming (simply stating “allotype nomenclature system” is not quite clear enough for me to follow).

*Many heavy chain mutations are described in Tables 1 + 2. The purpose of some of these mutations is discussed in the text, but it is not straightforward to find relevant discussions when looking at the tables. It would be helpful to make a separate table that describes the impact of the mutations in Tables 1 + 2 (otherwise perhaps Tables 1+2 could be modified to have some kind of key describing the impact of the described modifications)

*There are a few typographical/grammatical corrections that would improve the clarity of the text:

-Line 28 “add up to the many…” the word up should be deleted

-Line 52 ”is attributed why the…” why should be replaced with by

-Line 226-227 “The G4e1 mutation, which is not protected by a patent, is sufficient to prevent the Fab-arm exchange phenomenon (Figure 2) that has also been associated with in vitro instability” This sentence makes it sound like the mutations is associated with instability, which I don’t think is true. Should be rewritten

-Line 258 “antagonists antibodies?” should be “antagonist antibodies?”

-Line 388 “this observation does not militate in favor…” Should be mitigate instead of militate?

* In the section discussing anti-CTLA4 antibodies, this reference should be added https://doi.org/10.1073/pnas.1801524115
